# Human FBXL8 Is a Novel E3 Ligase Which Promotes BRCA Metastasis by Stimulating Pro-Tumorigenic Cytokines and Inhibiting Tumor Suppressors

**DOI:** 10.3390/cancers12082210

**Published:** 2020-08-07

**Authors:** Shu-Chun Chang, Wayne Hsu, Emily Chia-Yu Su, Chin-Sheng Hung, Jeak Ling Ding

**Affiliations:** 1The Ph.D. Program for Translational Medicine, College for Medical Science and Technology, Taipei Medical University, Taipei 110, Taiwan; sc.chang@tmu.edu.tw; 2Division of Acute Care Surgery, Department of Surgery, Taipei Medical University Hospital, Taipei 110, Taiwan; wayne.hsu131@gmail.com; 3Graduate Institute of Biomedical Informatics, College of Medical Science and Technology, Taipei Medical University, Taipei 110, Taiwan; emilysu@tmu.edu.tw; 4Clinical Big Data Research Center, Taipei Medical University Hospital, Taipei 110, Taiwan; 5Division of General Surgery, Department of Surgery, Shuang Ho Hospital, Taipei Medical University, Taipei 110, Taiwan; 6Division of Breast Surgery, Department of Surgery, Taipei Medical University Hospital, Taipei 110, Taiwan; 7Division of General Surgery, Department of Surgery, School of Medicine, College of Medicine, Taipei Medical University, Taipei 110, Taiwan; 8Department of Biological Sciences, National University of Singapore, Singapore 117543, Singapore; 9Graduate School of Integrative Sciences and Engineering, National University of Singapore, Singapore 119077, Singapore

**Keywords:** FBXL8 SCF E3 ubiquitin ligase, CCND2, IRF5, breast cancer (BRCA) and metastasis, pro-tumorigenic microenvironment

## Abstract

The initiation and progression of breast cancer (BRCA) is associated with inflammation and immune-overactivation, which is critically modulated by the E3 ubiquitin ligase. However, the underlying mechanisms and key factors involved in BRCA formation and disease advancement remains under-explored. By retrospective studies of BRCA patient tissues; and gene knockdown and gain/loss-of-function studies, we uncovered a novel E3 ligase, FBXL8, in BRCA. A signature expression profile of F-box factors that specifically target and degrade proteins involved in cell death/survival, was identified. FBXL8 emerged as a prominent member of the F-box factors. Ex vivo analysis of 1349 matched BRCA tissues indicated that FBXL8 promotes cell survival and tumorigenesis, and its level escalates with BRCA progression. Knockdown of FBXL8 caused: (i) intrinsic apoptosis, (ii) inhibition of cell migration and invasion, (iii) accumulation of two tumor-suppressors, CCND2 and IRF5, and (iv) downregulation of cancer-promoting cytokines/chemokines; all of which curtailed the tumor microenvironment and displayed potential to suppress cancer progression. Co-IP study suggests that two tumor-suppressors, CCND2 and IRF5 are part of the immune-complex of FBXL8. The protein levels of CCND2 and IRF5 inversely correlated with FBXL8 expression, implying that FBXL8 E3 ligase was associated with the degradation of CCND2 and IRF5. Altogether, we propose the exploitation of the ubiquitin signaling axis of FBXL8-CCND2-IRF5 for anti-cancer strategies and potential therapeutics.

## 1. Introduction

Breast cancer (BRCA) is the most common malignancy in women, being responsible for more than 500,000 deaths each year worldwide. In Western countries, BRCA constitutes 25% of female malignancies [1,2]. Increasing evidence suggest that the ubiquitin machinery plays a pivotal role in cancer development and progression [3,4]. However, the key factors and underlying mechanisms involved in BRCA are not fully explored. Recent findings on the involvement of ubiquitin proteasome system (UPS) in cancer biology is gathering significance in cancer therapy. For example, MLN4924 is a newly discovered investigational small molecule inhibitor of NEDD8 Activating Enzyme (NAE), currently under Phase I clinical trial (ClinicalTrials.gov Identifier: NCT03330106) with participants bearing advanced solid tumors. MLN4924 was reported to inactivate SCF (Skp1-cullin-F-box proteins) E3 ligase by inhibiting cullin neddylation [5]. Cumulative studies have demonstrated that MLN4924 inhibited cancer cell growth, including BRCA [6,7]. Both in vivo and in vitro studies showed that upon stimulation, the SCF E3 ubiquitin machinery can induce dynamic changes to a series of cytokine expression, which favors a pro-tumorigenic microenvironment [8,9]. Therefore, there may be vital members of the SCF E3 ubiquitin ligase family, which could drive pro-tumorigenesis. Being the largest family of E3 ubiquitin ligases, the SCF is composed of four components: SKP1, a cullin, an F-box protein, and RBX1 or RBX2. This multimeric complex is responsible for ubiquitination-mediated degradation of target proteins involved in various biological processes [10]. A well-regulated SCF E3 ubiquitin ligase maintains homeostasis in cell proliferation and genome stability, hence preventing carcinogenesis [10]. Bortezomib was the first-in-class proteasome inhibitor which served as a proof-of-principle that the proteasome degradation pathways are important as cancer targets. However, the lack of specificity of bortezomib caused off-target effects, hence the need for target specificity in proteasome-inhibitor related cancer therapeutics.

Previously, we demonstrated that Skp1-cullin-F-box proteins (SCF) E3 ubiquitin ligase critically balances immune-overactivation and pro-tumorigenesis [11,12], prompting us to hypothesize a vicious network of chronic inflammation-driven activation of SCF E3 ligase, which probably facilitates ubiquitination-mediated degradation of cancer-modulating factors and also perturbs the pro-tumorigenic cytokines. To test this hypothesis, we first identified the specific SCF E3 ligases and their potential pathobiological partners, and then explore their mechanisms in driving BRCA progression, with a view to understanding how the SCF E3 ubiquitin ligases might act as significant risk factors in cancer advancement [13]. Since the F-box is responsible for the functional specificity of SCF E3 tetrameric complex [14], we focused our attention on the expression profiles of F-box factors. Notably, there are 71 genes in the human “F-box gene family”, including: (1) 21 F-boxes grouped as leucine-rich repeat protein, (2) 11 F-boxes with WD repeat domain and (3) 39 F-boxes grouped as “others” [15,16,17].

By ex vivo characterization of the “signature profile” of F-box factors in patients’ BRCA tissues, we identified for the first time, a novel F-Box factor, FBXL8, which was significantly upregulated with cancer advancement. FBXL8 was found to interact with two tumor-suppressors, CCND2 (cyclin D2) and IRF5 (interferon regulatory factor 5). Knockdown of FBXL8 in BRCA cells resulted in the accumulation of CCND2 and IRF5, concomitant with cancer suppression. CCND2, a member of the D-type cyclins has been implicated in cell cycle regulation, differentiation and malignant transformation, and is seemingly inactivated in human cancers [18]. IRF5 is a transcription factor, purported to be a cancer-suppressor which regulates apoptosis and immune activation [19,20]. Our findings will open new avenues for translational research on E3 ubiquitin ligase-related regulation and intervention of cancer. We propose the manipulation of FBXL8-CCND2-IRF5 signaling axis to dampen the pro-tumorigenic microenvironment and curb metastasis.

## 2. Results

### 2.1. Clinicopathological Parameters of BRCA Patient Tissues Used For the Global RNA Sequencing

Clinical information of five representative BRCA patients with confirmed cancer staging, was obtained retrospectively from the medical records of BioBank, Taipei Medical University. The clinical stages of patients were selected based on tumor-node-metastasis (TNM): stage I (*n* = 1), stage II (*n* = 2) and stage III (*n* = 2). The age of the BRCA patients ranged from 38 to 69 years, with a mean of 59 years. The histology type was diagnosed as invasive lobular carcinoma (ILC) or invasive ductal carcinoma (IDC). BRCA resection pathology reports included primary tumor, categories and pathologic lymph node status. Figure 1A summarizes the clinicopathological parameters of the BRCA patients.

### 2.2. FBXL8 Is Significantly Upregulated in Human Primary BRCA Tissues

To explore the clinical significance of the key components of SCF multimeric complex which might respond to BRCA progression, and to identify the corresponding F-box factors involved, we performed an NGS-based RNA-Seq to profile the gene expression of the F-Box members in human primary breast carcinoma tissues. The five breast carcinoma tissues which have been clinically staged and characterized alongside corresponding normal counterpart tissues, were subjected to global mRNA profiling. Although FBXW7 and FBXO4 are recognized as tumor suppressors of human cancers [21,22], these two F-box factors did not show significant difference in their levels of expression in BRCA tissues, based on our RNA sequencing data. Nevertheless, we found that several SCF components were overexpressed in BRCA, including *SKP1* (S-phase kinase-associated protein 1), *SKP2*, *CUL1* (cullin 1), *FBXW7* (F-box/WD repeat-containing protein 7), *SAG* (Sensitive to apoptosis gene), *FBXO4* (F-box only protein 4) and *RBX1* (ring-box 1) (Appendix A). Red arrow (↑) highlights the upregulation of the identified F-box gene, *FBXL8* (Figure 1A). Compared to normal control tissues, the mRNA level of FBXL8 in carcinoma tissues was significantly upregulated by up to 7.5-fold (*, False Discovery Rate, FDR < 0.05). Scatter plot (Figure 1B) illustrates quantitative differences in the expression levels of *FBXL8* mRNA in individual clinical samples, with 3-fold increase in breast carcinoma (expressed in log2). We found that amongst all 71 F-box family members, *FBXL8*, a novel F-box factor, was prominently upregulated in breast carcinoma tissues, compared to the corresponding paired normal tissues (FDR < 0.05).

Consistently, RNA-Sequencing data in The Cancer Genome Atlas (TCGA) showed that the mRNA expression of *FBXL8* was significantly upregulated in BRCA tissues (Figure 1C, ** *p* < 0.01). Since information on FBXL8 is lacking in literature, we performed sequence alignment of FBXL8 with that of several mammalian species using Clustal Omega and found that FBXL8 contains highly conserved amino acid sequence amongst mammals (Figure 1D), suggesting that it may play a significant biological role.

### 2.3. FBXL8 Is Upregulated with the Advancement of BRCA Stages

To affirm the RNA-Seq data (Figure 1) and to further understand the correlation between FBXL8 expression and the clinical status of BRCA patients, we next performed qRT-PCR and IHC staining with larger numbers of primary tissue samples (Figure 2). The *FBXL8* mRNA profiles of 32 paired samples (*n* = 64) were determined. Appendix A shows the clinicopathological parameters of the BRCA patient breast tissue samples used for qRT-PCR analysis. Consistently, the mRNA levels of *FBXL8* were significantly elevated (*p* < 0.01) in primary BRCA tissues (Figure 2A). The expression levels of *FBXL8* correlated significantly (*p* < 0.01) with clinical staging. In addition, *SAG* (sensitive to apoptosis gene), another important component of SCF E3 ubiquitin ligase, which confers anti-apoptosis and promotes liver cancer [12,23,24], was also significantly upregulated (*p* < 0.01) in all stages of BRCA (Figure 2B).

This further corroborates the critical role of the members of SCF E3 ubiquitin ligase in BRCA progression. IHC of FBXL8 in 30 paired BRCA tissues (*n* = 60) also showed increase in FBXL8 with disease progression from stages IA to IIA/B to IIIA/B/C (Figure 2C). The clinicopathological parameters of the corresponding tissue sections are shown in Appendix A. The expression of both the mRNA and protein of FBXL8 in BRCA tissues were consistently increased by up to 4-fold compared to normal tissues, and the levels in BRCA tissues were found to elevate with the advancement of cancer (*p* < 0.001) (Figure 2D). Clearly, the FBXL8, a newly identified F-box factor is associated with the progression of BRCA. The rise in the ubiquitin protein in BRCA tissues (Figure 2E) is possibly associated with a significant role of the UPS in BRCA development. Altogether, we have demonstrated simultaneous increases in the levels of FBXL8, SAG and ubiquitin, all of which are functionally related to the UPS-mediated protein degradation system, in BRCA formation.

### 2.4. FBXL8 is a Novel Anti-Apoptosis Factor in BRCA

To elucidate the mechanisms underlying the pathophysiological significance of FBXL8, we performed gene specific-siRNA knockdown of FBXL8 in breast carcinoma cells to assess the potential loss/gain-of-function. We first examined the endogenous levels of FBXL8 in two human BRCA cell lines, including a highly invasive MDA-MB231, a less invasive MCF7, and a control cell line, MCF10A which expresses very low basal level of FBXL8. Prior to knockdown, we observed 23- and 15- folds higher expression of FBXL8 in MCF7 and MDA-MB231, respectively, compared to that of the control MCF10A cells (*p* < 0.01) (Figure 3A). Consistent with mRNA profiles, we observed only minimal expression of FBXL8 protein in MCF10A cells (Appendix A). Henceforth, for knockdown analysis, we focused our attention on MCF7 and MDA-MB231 cells. With up to 95% knockdown efficiency of FBXL8 in MCF7 and MDA-MB231 cells (Figure 3B), we noted a reduction in cell viability by 52.5% within 48 h (an average taken from MTT and CTB assays) (Figure 3C,D), which was significantly different (*p* < 0.001) from that of control siRNA-treated cells.

FBXL8-specific siRNA knockdown effected 2–3-fold reduction in cell proliferation of both MCF7 and MDA-MB231 cells (Figure 3E,F, *p* < 0.001). This reduction in cell proliferation is supported by significant increases in early apoptosis of 20% and 24% in FBXL8-knocked down MCF7 and MDA-MB231 cells, respectively (Figure 3G). Representative histograms of apoptosis assay are shown in Appendix A (*p* < 0.001). Further investigation of Caspase activity showed that both Caspases-9 and -3 were activated in FBXL8 knocked down cells (Figure 3H), indicating that the presence/increase in FBXL8 prevents intrinsic apoptosis and confers survival advantage to BRCA, hence favoring cancer progression.

### 2.5. Knockdown of FBXL8 Inhibited BRCA Cell Migration and Invasion

To further investigate the impact of FBXL8 as a driver of BRCA metastasis, we examined the effects of FBXL8-knockdown on cell migration and invasion. When monitored over 30 h, the highly invasive MDA-MB231 cells showed 40% slowdown in cell migration compared to 14% with the less invasive MCF7 (both compared to control siRNA-treated cells, *p* < 0.001) (Figure 4A–D). Consistently, cell invasion was suppressed by 4–8 fold in MDA-MB231 and MCF7 cells (*p* < 0.001) (Figure 4E, Appendix A). Importantly, direct analysis of cell growth showed that the doubling time for MCF7 and MDA-MB231 cells was 26 h and 23 h, respectively (Figure 3D). The 4–8 fold reduction in invasion observed at 24 h therefore cannot be solely attributable to the inhibition of cell growth or a difference in cell doubling time. These results again support that FBXL8 is a significant contributor to BRCA cell growth and metastasis.

### 2.6. Knockdown of FBXL8 Suppressed Pro-Inflammatory Cytokines/Chemokines in BRCA

There is increasing evidence that the SCF E3 ubiquitin machinery favors cancer formation by controlling immunomodulators such as cytokines which are abundant in the tumor microenvironment [8,9,25]. It was therefore pertinent for us to examine whether and how FBXL8 might regulate the cytokines. A protein array analysis of FBXL8-knocked down BRCA cell lysate showed that the loss of FBXL8 caused a downregulation of the pro-tumorigenic factors, including MCP-1, I-TAC, TECK, CTACK, MIF, GM-CSF, NT-3, FGF-6, angiogenin, ICAM-1, DtK and EGFR (Figure 5A). Interestingly, we observed that angiogenin, GM-CSF, ICAM-1 and EGFR were dramatically over-expressed (for example, up to 20-fold increase for EGFR) in the highly invasive MDA-MB231 cells compared to the less invasive MCF7 cells. Previous findings have also shown that these factors support the survival of breast cancer cells and promote angiogenesis, further triggering cancer invasion and metastatic spread [26,27,28,29]. Additionally, NT-3, which was concomitantly decreased by knockdown of FBXL8, has been reported to modulate the tumor microenvironment to promote breast-to-brain metastasis [30]. Suffice to say, knockdown of the pro-tumorigenic FBXL8 caused the loss of another pro-tumorigenic factor, NT-3, which plausibly further restrains BRCA metastasis. It would be interesting, in future, to examine the potential functional partnership between FBXL8 and NT-3, in their co-regulation of BRCA progression.

Quantification by ELISA further affirmed that depletion of FBXL8 significantly reduced the production of several cancer-promoting factors e.g., GM-CSF, MCP-1, CTACK, EGFR, ICAM-1, I-TAC, MIF and TECK (Figure 5B, blue bars). Such factors are reported to be associated with cancer survival, growth and/or metastasis in BRCA [27,31,32,33,34,35,36,37,38]. The association of these factors to FBXL8 activity suggests that FBXL8 promotes BRCA survival by modulating the cytokine/chemokine-rich pro-tumorigenic microenvironment.

On the other hand, osteoprotegerin (OPG, overexpressed in BRCA metastasis) is up-regulated in MCF7 and down-regulated in MDA-MB231 during FBXL8 knockdown. Although we found opposite trends of OPG in the two BRCA cell types, higher levels of OPG have been associated with an increased risk of ER-negative BRCA [39]. Similarly, knockdown of FBXL8 increased the expression of pro-tumorigenic TNFR1. Inhibition of TNFR1 expression was reported to increase apoptosis, and inhibit cell proliferation and invasion in triple-negative and ER positive BRCA cells [40]. IL-6R was negatively regulated by FBXL8 expression, although IL-6 appeared unaffected by FBXL8 knockdown. Such data highlight the complexity of FBXL8-associated immune network. It is plausible that the BRCA tissues contained heterogeneous tumor cell types. Therefore, our findings suggest the propensity for developing variant forms of FBXL8-based combinatorial treatments, possibly with (i) FBXL8-mediated cytokines or (ii) FBXL8-binding partners, for specific and targeted chemotherapy against different heterogeneous forms of BRCA. This prompted us to further search for and characterize FBXL8-binding partners.

### 2.7. Cancer-Suppressors, CCND2 and IRF5, Are FBXL8-Binding Partners in BRCA

FBXL8 is a leucine-rich repeat protein member of the E3 SCF multimeric complex. Our in silico modeling prediction revealed FBXL8-binding partners including CUL1, SKP1, CUL2 and CUL7 (Appendix A). These results support the involvement of E3 ligase in promoting BRCA. In addition, several non-SCF E3 ligase-associated components were proposed, including cell-cycle associated proteins, cancer-promoting transcription factors and DNA repair/replication factors. Notably, among the predicted list of FBXL8-binding factors, two cancer suppressors, Cyclin D2 (CCND2) and Interferon Regulatory Factor 5 (IRF5), emerged prominently as potential interaction partners of FBXL8. We envisage that the physiological functions of CCND2 and IRF5 would oppose the pro-tumorigenic role of FBXL8. While FBXL8 promotes cancer growth/metastasis, CCND2 and IRF5 suppress cancer progression. Knockdown of CCND2 has been reported to increase the growth rate and migration ability of cancer cells [41]. On the other hand, IRF5 overexpression is capable of reverting MDA-MB231 to normal acini-like structures [20]. Furthermore, our IHC staining of breast carcinoma tissues demonstrated a reciprocal profile of FBXL8 versus CCND2 and IRF5 with cancer advancement over stages I–III. The FBXL8 level increased while CCND2 and IRF5 levels decreased (Figure 6A–C). The mRNA levels of CCND2 and IRF5 were further examined by both NGS-based RNA-Seq and TCGA database analyses. Figure 6D–G showed that the IRF5 mRNA was significantly upregulated in breast tumor tissues (Figure 6G), although its protein level was reduced in BRCA tissues.

This suggests potential post-translational modifications which led to the loss of IRF5 proteins in BRCA. Altogether, our computer modeling prediction and retrospective studies highlight the potential role of FBXL8 E3 ligase in specific degradation of CCND2 and IRF5 proteins as cancer progresses.

We next investigated whether anti-cancer CCND2 and IRF5 are binding partners of FBXL8, with a view to providing explanations on how their interactions might impact BRCA progression. To test this hypothesis, protein extracts of MCF7 and MDA-MB231 cells were first verified to contain IRF5, CCND2 and FBXL8 (Appendix A). Co-immunoprecipitation (Co-IP) showed that FBXL8 pulled down both CCND2 and IRF5 in MCF-7 and MDA-MB231 cells (Figure 6H, Appendix A), suggesting that CCND2 and IRF5 are part of the immune-complex of FBXL8. IP with control IgG_2b_ demonstrated the specificity of the assay. Importantly, while the mRNA levels of CCND2 and IRF5 were maintained in FBXL8-knockdown cells (Appendix A), we observed a sustained rise in the protein levels of CCND2 and IRF5 (Figure 6I, Appendix A), suggesting that the accumulation of CCND2 and IRF5 proteins was attributable to a posttranscriptional event. The knockdown of FBXL8 caused the accumulation of CCND2 and IRF5 specifically (but not tubulin) proteins in both BRCA cell lines, suggesting that FBXL8 selectively regulates the protein degradation of CCND2 and IRF5 (Figure 6I), thus explaining why FBXL8 rises while CCND2 and IRF5 declines as BRCA progresses. Given the critical role of CCND2 and IRF5 in cell cycle regulation as growth-inhibitors [18,20], our data here account for the functional significance of FBXL8 as a tumor-promoter, whose knockdown spared the tumor suppressors, CCND2 and IRF5 from degradation, thus allowing them to accumulate and suppress BRCA progression.

## 3. Discussion

Due to the lack of early symptoms and a dearth of markers for dynamic progression of BRCA, breast cancer malignancy is often detected too late. BRCA is associated with inflammation and immune-overactivation conditions, which perpetuates cancer recurrence and lowers patient survival rate [13]. Concordant with our earlier findings that showed the involvement of SCF E3 in inflammation and tumorigenesis [11], treatment with non-steroidal anti-inflammatory drugs has been shown to reduce BRCA metastasis and recurrence of the disease [42,43]. Other clinical efforts to arrest BRCA have used bortezomib (Velcade or PS-341), the first proteasome inhibitor to be approved by FDA. Unfortunately, bortezomib was found to be cytotoxic due to its inhibition of protein degradation on a global scale [44]. Therefore, we reasoned that the ideal approach is to target a specific protein degradation enzyme like an E3 ligase, where different F-box factor family members are selectively activated in a disease-specific manner [45]. This approach should improve the specificity of anti-BRCA targeting. Here, by retrospective studies with BRCA patient tissues (*n* = 134 from Taiwan BioBank and *n* = 1215 from TCGA database), we report that FBXL8, a novel F-box E3 ligase family member, is significantly correlated with BRCA disease progression. For the first time, we revealed the pathobiological functions of FBXL8 and elucidated its functional significance ex vivo. We found that FBXL8 blocks apoptosis of BRCA cells and modulates cytokine profiles to promote a pro-inflammatory condition. Furthermore, we showed that knockdown of FBXL8 induced intrinsic apoptosis of the BRCA cells, suggesting that FBXL8 promotes BRCA advancement, as supported by the involvement of other F-box proteins in regulating the EMT process in cancers [14]. Consistently, we demonstrated that the loss of FBXL8 decreased cell migration and invasion, affirming that FBXL8 is a key promoter of BRCA progression and metastasis. This is strongly corroborated by the increase in FBXL8 level as the disease advanced in BRCA patients (Figure 2; **, *p* < 0.01; ***, *p* < 0.001). Congruent to these findings, knockdown of FBXL8 significantly down-regulated cancer-promoting cytokines and chemokines. Therefore, repression of FBXL8 is a potential anticancer strategy to activate apoptosis, to inhibit metastasis, and to modulate pro-tumorigenic cytokines in BRCA (Figure 7A).

Further insights on the translational impact of FBXL8 was accrued from in silico modeling prediction, which revealed various potential interaction partners of FBXL8, for example: Cyclin family members like cancer suppressor CCND2 [18], IRF5 and DNA repair/replication-associated proteins (nibrin and SMAD4, Appendix A). Thus, FBXL8 appears rather promiscuous, associating with multiple binding partners, to plausibly promote cancer formation and progression through the turnover of: (i) cell-cycle associated proteins, (ii) cancer- suppressors and (iii) proteins responsible for DNA repair/replication. We further validated two of the in silico predicted binding partners, CCND2 and IRF5 (tumor-suppressors), and empirically demonstrated their functional roles which might countermeasure FBXL8, in cancer cells. Notably, the specificity of the UPS in protein degradation is mainly controlled at the level of E3 ligase [46], and it is possible that CCND2 and IRF5 are the specific substrates targeted for degradation by FBXL8, an E3 ligase. The molecular axes of interactions of FBXL8-CCND2, FBXL8-IRF5 and/or FBXL8-CCND2-IRF5 prompted us to speculate that their partnership(s) could fine-tune and maintain cellular homeostasis (Figure 7B). Should the balance of this partnership tip over in favor of FBXL8, the tumorigenic potential will rise. Since IRF-5 has been reported to stimulate inflammatory genes [20,47], the involvement of IRF5 indicates how FBXL8 might indirectly influence the cytokine profiles, hence promoting a pro-inflammatory cytokine-rich cancer microenvironment, as our results alluded to (Figure 5). Additionally, since IRF5 itself can also act as a pro-tumorigenic factor [48], it is plausible that besides E3, different E2s in the UPS may also respond to the pathophysiological conditions to modulate the FBXL8-IRF5 axis, which should enhance the tumorigenic and metastatic potential.

Since FBXL8-knockdown raised the levels of CCND2 and IRF5, which resulted in cancer suppression, it is conceivable that as a tumor-promoter, FBXL8 acts as a dominant member of a tripartite liaison, in which FBXL8 controls the protein level and functionality of the two tumor suppressors, CCND2 and IRF5 (Figure 6). Of significance is that FBXL8 is a specific protein degradation enzyme which selectively targets cancer suppressors like CCND2 and IRF5. Unlike Bortezomib which is a general protein degradation enzyme, we propose that FBXL8-blockers can be developed to specifically unleash CCDN2 and IRF5, in BRCA treatment. Furthermore, antibodies against the pro-inflammatory cytokines identified in this study may be employed clinically in combination with FBXL8-blocker to treat heterogeneous variants of BRCA.

## 4. Materials and Methods

### 4.1. Tissue Samples

Sixty seven pairs of tissue samples (tumor and normal tissues; *n* = 134) from BRCA patients were acquired from the BioBank (Taipei Medical University, Taipei, Taiwan), after clinical diagnosis was confirmed by biopsy and histological evaluation. All patient samples were collected with informed consent. The study was approved by the TMU-JIRB (Taipei Medical University-Joint Institutional Review Board; IRB: N201803107). Adjacent non-cancerous breast tissues were obtained at least 2 cm away from the tumor node, to serve as paired normal tissue control. Ex vivo experiments included (i) NGS-based RNA-seq analysis, where five-pairs of the primary tissues (*n* = 10) were obtained from: one stage I, three stage II and one stage III patient samples (the corresponding clinicopathological information are shown in Figure 1A); (ii) qRT-PCR analysis of 32 paired tissues (*n* = 64), including fifteen stage I, nine stage II and eight stage III patient samples (the corresponding clinicopathological information are shown in Appendix A); and (iii) immunofluorescence staining, where 30 pairs of the tissues (*n* = 60) were used, including: nine stage I, eleven stage II and ten stage III patient samples (the corresponding clinicopathological information are shown in Appendix A). H&E (hematoxylin and eosin) sections were obtained from the Taipei Medical University BioBank. For immunofluorescence staining, paraffin-embedded tissues were sectioned at 5 μm thickness, dehydrated and blocked in Ultravision Protein Block (TA060PBQ, Thermo Scientific, Dublin, Ireland) for 10 min at room temperature. Then, the sections were incubated with primary and secondary antibodies overnight and 1 h, respectively. The stained tissue sections were observed under a microscope (Olympus, Tokyo, Japan) and images were acquired using the EOS Utility software (version 3.10.0) (Canon, Uxbridge, UK).

### 4.2. Cell Lines and Reagents

Human BRCA cell lines (MCF7, MDA-MB231) and an FBXL8-negative control breast cell line (MCF10A) were obtained from American Type Culture Collection (ATCC, Gaithersburg, Maryland, USA). MCF7 and MDA-MB231 were cultured in complete DMEM medium (Gibco, Paisley, Scotland, UK). MCF10A was cultured in complete DMEM/F12 medium (Gibco). All the complete media were supplemented with 10% FBS (Thermo Scientific) and 100 U/mL penicillin and 100 μg/mL streptomycin (Invitrogen, Carlsbad, CA, USA). Antibodies used in IHC (immunohistochemistry), immune blotting and immunoprecipitation analysis were anti-FBXL8 antibody (sc-390582, Santa Cruz Biotechnology, Dallas, TX, USA), anti-ubiquitin antibody (ab7780, Abcam, Cambridge, UK), anti-CCND2 antibody (ab226972, Abcam), anti-IRF5 antibody (ab2932, Abcam), normal mouse IgG2b (sc-3879, Santa Cruz Biotechnology) and anti-Tubulin antibody (ab176560, Abcam).

### 4.3. Next-Generation Sequencing (NGS)-Based RNA-Seq Analysis

#### 4.3.1. RNA Isolation for NGS-Based RNA-Seq

Total RNA was extracted by Trizol^®^ Reagent (Invitrogen) according to the instruction manual. Purified RNA was quantified at OD_260nm_ by using Qubit (Qubit^®^ 2.0 Fluorometer, Life Technologies, Singapore, Singapore) and analysed using a bioanalyzer (Bioanalyzer 2100 system, Agilent, Santa Clara, CA, USA) with RNA 6000 labchip kit (Agilent Technologies, Santa Clara, CA, USA).

#### 4.3.2. Library Preparation & Sequencing for NGS-Based RNA-Seq Application

All procedures were carried out according to the manufacturer’s protocol from Illumina (San Diego, CA, USA). Library construction of all samples was performed using Agilent’s SureSelect Strand Specific RNA Library Preparation Kit (100SE, Single-End) bp sequencing on a Solexa platform. The sequences were determined using sequencing-by-synthesis technology via the TruSeq SBS Kit. Raw sequences were obtained from the Illumina Pipeline software bcl2fastq v2.0, which was expected to generate 5M (million reads) per sample.

#### 4.3.3. RNA Sequence Analysis

Quality control of raw sequence data was performed with FastQC. Adapter and quality trimming was performed using cutadapt (phred cutoff: 20; minimum length: 25) [49]. Reads were mapped to the human reference genome using STAR aligner indexed with GRCh37 assembly and GENCODE Release 26 v26lift37 annotation. Gene expression was quantified using RSEM. The expected counts were used as input for the differential gene expression analysis between tumor and adjacent-normal BRCA samples using edgeR in the R environment. A false discovery rate (FDR) < 0.05 was used as a threshold to define genes that showed statistically significant differential expression. A log_2_ fold change (logFC) less than 0 indicates under-expression in the tumor samples, whereas logFC greater than 0 represents over-expression in the tumor samples.

### 4.4. TCGA Database Analysis

TCGA breast cancer transcript dataset was downloaded from R package, TCGAbiolinks [50]. In total, 1215 primary tissues were analyzed, including 1102 carcinoma tissues and 113 normal tissues. All of the raw counts were used as input for the differential gene expression analysis between tumor and normal breast samples, using DESeq2. The *p* adjust value < 0.05 was used as threshold to define genes that showed statistically significant differential expression.

### 4.5. Quantitative Real-Time PCR

Total RNA was extracted with TRIzol reagent (Invitrogen) and the RNA was reverse transcribed using random hexamers and HiSenScript™ RH(-) RT-PCR PreMix Kit (iNtRON Biotechnology, Seongnam, South Korea), according to the manufacturer’s instructions. The synthesized cDNA was used for real-time PCR analysis using SYBR Green (Life Technologies Carlsbad, CA, USA) on a QuantStudio 3 Real-Time PCR System (Applied Biosystems, Foster City, CA, USA). The primers were human FBXL8 (83 bp product): sense, 5′-AATCAGTTGCGAATGTGAGC-3′; antisense, 5′-CCAGCCGTAGGTTGTGA ATG-3′; human SAG (109 bp product): sense, 5′-CAGGCTCCAAGTCGGGAGGCG-3′, antisense, 5′-TGGACCCTGCAGATGGCACAGG-3′; human CCND2 (118 bp product): sense, 5′- CACTTGTGATG CCCTGACTG-3′; antisense, 5′-ACGGTACTGCTGCAGGCTAT-3′; human IRF5 (141 bp product): sense, 5′-ATGCTGCCTCTGACCGA-3′; antisense, 5′- GCCGAAGAGTTCCACCTG-3′; human GAPDH (131 bp product): sense, 5′-GTCTCCTCTGACTTCAACAGCG-3′, antisense, 5′-ACCAC CCTGTTGCTGTAGCCAA-3′. All expression values were normalized based on GAPDH as an endogenous control.

### 4.6. FBXL8 Knockdown in BRCA Cells

FBXL8 siRNA (siGENOME Human FBXL8-SMARTpool, M-017504-00-0005) was purchased from Dharmacon (Lafayette, CO, USA). Control (scrambled) siRNA were purchased from Invitrogen. siGENOME Human FBXL8-SMARTpool contains a mixture of 4 siRNAs provided as a single reagent; providing advantages in both potency and specificity, which was anticipated to achieve silencing effects > 75%. siRNA transfection into MCF7 or MDA-MB231 cells (at 5 × 10^5^ cells per well of a 6-well plate) was conducted using 7.5 μL TransIT-X2 Transfection Reagent (Mirus, Madison, WI, USA) with 25 nM of siRNAs per well.

### 4.7. Cell Viability Assay

Both 3-(4,5-dimethylathiazol-2-yl)-2,5-diphenyl tetrazolium bromide (MTT) and CellTiter Blue (CTB) obtained from Life Technologies (Carlsbad, CA, USA) and Promega (Madison, WI, USA), respectively, were used to determine the cell viability, according to the manufacturer’s instructions. To assess the cell viability the functional mitochondrial activity was measured by incubating the cells with either 10 μL MTT reagent in 100 μL culture medium at 37 °C for 4 h, or 10 μL CTB reagent at 37 °C for 4 h. The metabolic activity of viable cells was measured at 570 nm using a Synergy H4 microplate reader (BioTek, Winooski, VT, USA) at time points indicated. Samples from each time point were normalized with corresponding PBS (phosphate buffered saline) controls.

### 4.8. Cell Proliferation Assay

After cells are transfected with FBXL8-specific siRNA or control scrambled siRNA for 16 h, both Alamar blue and Trypan Blue dye exclusion were used to measure the cell proliferation. The cell growth was measured by Alamar blue. The metabolic activity of the cells was determined at 570 nm using a Synergy H4 microplate reader (BioTek) at time points indicated. For Trypan Blue dye exclusion, to assess total cell number, cells were scraped and resuspended in equal volumes of culture medium and trypan blue dye (0.4% solution; Gibco, Waltham, MA, USA) and counted using an improved Neubauer hemocytometer.

### 4.9. Cellular Apoptosis Assay

MCF7 and MDA-MB231 cells were transfected with FBXL8-specific siRNA or control scrambled siRNA for 24 h, followed by determination of apoptosis. Early apoptosis was measured using annexin V (BioLegend, San Diego, CA, USA) in conjunction with 7-AAD (BioLegend), according to manufacturer’s instructions. Annexin V identifies surface-exposed phosphatidylserine, and 7-AAD is retained in late apoptotic cells. Early apoptosis (Annexin V^+^/7-AAD^−^) was then analyzed using FACScan flow cytometer (FACSVerse, BD Biosciences, Franklin Lakes, NJ, USA) for a minimum of 10,000 events.

### 4.10. Caspases-9 and -3 Assays

24 h after the transfection of FBXL8-specific siRNA or control scrambled siRNA, apoptosis of MCF7 and MDA-MB231 cells were examined by determining the caspase-specific cleavages of activated caspases-9 and -3 (Abcam). The p-nitroanilide (p-NA) light emission was measured at 405 nm using a Synergy H4 microplate reader (BioTek).

### 4.11. Cell Migration and Invasion Assays

24 h after treatment of the MCF7 and MDA-MB231 cells with FBXL8-siRNA or controls, cell migration assay was performed. For collection, the cells were rinsed with PBS, and harvested using 0.05% Trypsin-EDTA (Invitrogen). The cells were then plated into a 2-well Culture-Insert (ibidi, Martinsried, Germany) according to the manufacturer’s instructions. The Culture-Inserts were removed after 16 h, and cell migration was monitored over the indicated time-scale. For cell invasion assay, biocoat Matrigel invasion chambers with 8-μm pores in 24-well plates (Corning, Discovery Labware, Inc., Bedford, MA, USA) were used. 24 h after siRNA transfection, the cells were plated onto the chambers. The cells were detached with 0.05% Trypsin-EDTA, resuspended in conditioned medium (10% FBS (fetal bovine serum)) and added to the upper compartment of the chambers, according to the manufacturer’s instructions. After 24 h of incubation at 37 °C, the cells on the upper surface were completely removed by wiping with a cotton swab, and then the filter was fixed with 100% methanol and stained with crystal violet solution (0.5% (*w/v*) crystal violet in 25% (*v/v*, methanol). Cells that had migrated from the upper to the lower side of the filter were imaged and counted with a light microscope (five fields/filter).

### 4.12. Cytokine Arrays

The secretion of cytokines from BRCA cells effected by FBXL8-knockdown was analyzed using human cytokine antibody array (human cytokine antibody array C6 and C7, AAH-CYT-1000) purchased from RayBiotech (Peachtree Corners, GA, USA). Cells were cultured in complete medium with transfection of FBXL8-specific siRNA or control scrambled siRNA for 48 h. Culture supernatants were collected for cytokine antibody array analysis. Cytokine signal intensities were quantified by Scion Image software (Frederick, MD, USA). The fold-change of cytokines secreted from cells was calculated by normalization of FBXL8-specific siRNA-treated samples to control scrambled siRNA-treated samples.

### 4.13. ELISA Quantification of Cytokines

To determine the effects of FBXL8 on cytokine release, both MCF7 and MDA-MB231 were transiently transfected with FBXL8-specific siRNA or control siRNA for 48 h. The levels of MCP-1, EGFR, I-TAC, TECK, CTACK, MIF, GM-CSF and ICAM-1 (based on the cytokine array results), secreted by the cells, were quantified in supernatants derived from 48-h cultures, by using the respective ELISA (R&D Systems, Minneapolis, MN, USA), according to manufacturer’s instructions.

### 4.14. Co-IP and Immunodetection

48 h after treatment of the MCF7 and MDA-MB231 cells with FBXL8-siRNA or control siRNAs, Co-IP assay was performed. For Co-IP study with MCF7 or MDA-MB231 total cell lysates, the cells were lysed in CHAPS lysis buffer containing 20 mM Tris-HCl, pH 7.5, 5 mM MgCl2, 137 mM KCl, 1 mM EDTA, 1 mM EGTA, 1% CHAPS and 1 protease inhibitors (complete EDTA-free cocktail, Roche, St. Louis, MO, USA). Then, the cell lysate was precleared by incubating with protein G Sepharose (GE Healthcare, Little Chalfont, UK) at 4 °C for 2 h. The supernatant was incubated overnight at 4 °C with FBXL8 or control IgG_2b_ antibody, followed by a 3-h incubation with protein G Sepharose. The washed immunoprecipitates, resuspended in Laemmli buffer, was boiled at 95 °C for 5 min before immunodetection of FBXL8, CCND2 or IRF5.

### 4.15. Statistical Analysis

Data were expressed as means ± S.D. from three independent experiments, with three replicates per sample/condition tested. Differences between averages were analyzed by two-tailed Student’s t-test. Significance was set at *p*-value of <0.01 (**, *p* < 0.01; ***, *p* < 0.001). The acquired data from FACS were analyzed with Summit software (Version 4.3.02) (Fort Wayne, IN, USA). The relative migration rate, indicated as % gap closure, was calculated using Image J analysis software (version ImageJ2). All target signals from IHC were quantified by HistoQuest software (version 7.0) (TissueGnostics, Vienna, Austria).

## 5. Conclusions

Here, based on global analysis of NGS-sequenced transcripts of BRCA patient tissues (*n* = 1349), we identified FBXL8 to be a novel and potent promoter of BRCA. We demonstrated for the first time that knockdown of FBXL8 in BRCA cells: (i) suppressed cell survival via elevating intrinsic apoptosis, (ii) decreased cell migration and invasion, (iii) caused accumulation of tumor-suppressors, CCND2 and IRF5 and (iv) downregulated cancer-promoting cytokines and chemokines, hence curtailing the tumor microenvironment. We showed a novel molecular mechanism underlying the specific interaction of the pro-tumorigenic FBXL8 with two cancer suppressors, CCND2 and IRF5, resulting in their downregulation. We propose the translational potential in modulating the partnership axes of FBXL8-CCND2-IRF5 to target BRCA.

## Figures and Tables

**Figure 1 cancers-12-02210-f001:**
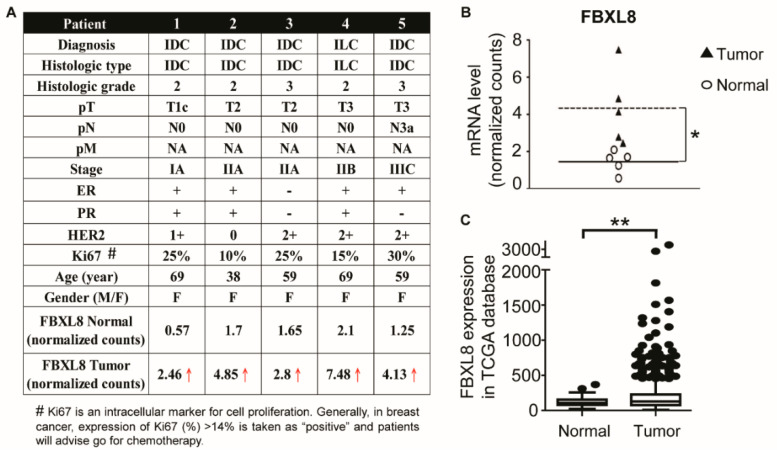
The *FBXL8* mRNA level is significantly upregulated in human breast carcinoma tissues. To characterize the expression profiles of F-box family members in human breast cancer (BRCA), a retrospective study was performed by NGS-based RNA-Seq. Breast carcinoma tissues and the corresponding normal counterparts were characterized for their global mRNA profiles. (**A**) Clinicopathological parameters of five representative BRCA patient carcinoma tissue samples were studied. Red arrow highlights the upregulation of *FBXL8*. Clinicopathological information provided from medical records includes age, gender (F, Female), pT (Primary Tumour), pN (pathologic lymph node status), pM (Distant Metastasis), etiology and staging of the patients by clinical tumor-node-metastasis. IDC, Invasive ductal carcinoma; ILC, Invasive lobular carcinoma; ER, estrogen receptor; PR, progesterone receptor. (**B**) Quantitative results from each patient. Each data point represents an EdgeR normalized count for *FBXL8* (*, FDR < 0.05). (**C**) Box-whisker plot of the *FBXL8* mRNA in TCGA breast cancer transcript database. Vertical lines, box and horizontal white line correspond to min-max range, 25–75th percentile range and median, respectively. Data are reported as normalized count as provided in the TCGA level 3 data (**, *p* < 0.01). (**D**) The sequence alignment of FBXL8 in mammalian species. Comparative amino acid sequence alignment of FBXL8 from human (NP_060848.2), chimpanzee (XP_511023.2), pig (XP_020949705.1), mouse (NP_056636.2), rat (NP_001102598.1), bovine (NP_001070405.1) and beaver (XP_020031370.1) shows conservation of residues among all species (color highlights). Clustal Omega software (version 1.2.4) was used for multiple sequence alignment.

**Figure 2 cancers-12-02210-f002:**
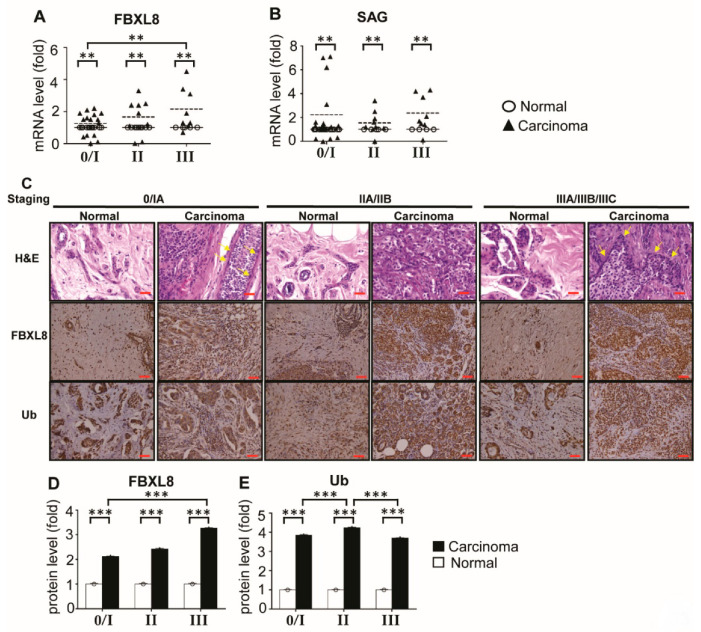
FBXL8 and key ubiquitin components are upregulated in human primary BRCA tissues. To clarify the potential of FBXL8-associated ubiquitination, analyses were performed by both qRT-PCR (A and B) and IHC (C). In total, 32 paired primary tissues are examined by qRT-PCR (*n* = 64). (**A**) FBXL8 shows significant overexpression in carcinoma tissues, particularly in advanced stages (**, *p* < 0.01). (**B**) *SAG* (sensitive to apoptosis gene), a key component of SCF E3 ubiquitin ligase, was also elevated in BRCA tissues throughout all disease stages. FBXL8 or SAG gene was normalized against GAPDH housekeeping gene. Data are representative of means ± SD (*n* = 3). **, *p* < 0.01; ***, *p* < 0.001. (**C**) H&E (hematoxylin and eosin) and immunohistochemistry (IHC) staining. Inflammatory cell infiltration is indicated with yellow arrows. Protein expression levels of FBXL8 and ubiquitin are examined in both normal (*n* = 30) and carcinoma (*n* = 30) breast tissues. The corresponding quantitative results of FBXL8 and ubiquitin in IHC are shown in panels (**D**) and (**E**), respectively. Consistent with the mRNA levels, the FBXL8 proteins are significantly upregulated in breast carcinoma tissues, as the cancer advanced into the later stages. Ubiquitin is also upregulated significantly in carcinoma tissues. The corresponding clinicopathological information for qRT-PCR analysis and IHC are shown in Appendix A and Appendix A, respectively. Brown color indicates DAB dye-stained proteins of interest. Scale bar is 100 μm, shown as the red color line (—). ***, *p* < 0.001.

**Figure 3 cancers-12-02210-f003:**
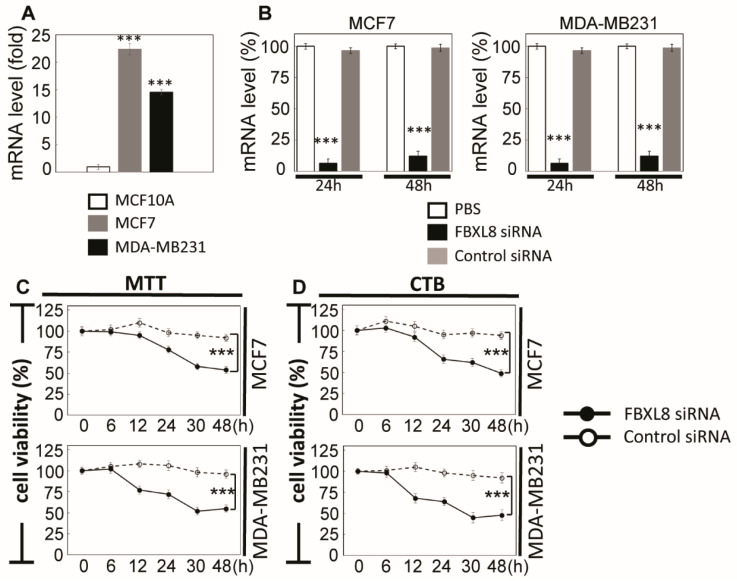
FBXL8 plays a critical anti-apoptotic role in BRCA. To elucidate the functional role of FBXL8 in breast carcinoma, we analyzed various human breast cancer cell lines: MCF-7 and MDA-MB231, compared to a non-cancer control breast cell line, MCF10A. (**A**) qRT-PCR shows the endogenous levels of FBXL8 mRNA in MCF7 and MDA-MB231, compared to MCF10A cells. FBXL8 mRNA is significantly upregulated in breast carcinoma cells. (**B**) qRT-PCR analysis was used to measure the RNAi efficacy. Optimized transfection efficacy with FBXL8-specific siRNA is shown in both MCF7 and MBA-MD231 cells. Knockdown efficiency was optimal at 24 h where, up to 95% loss of FBXL8 mRNA was achieved in both cell lines, compared to controls using PBS (phosphate buffered saline) and scrambled siRNA (***, *p* < 0.001). The cell viability of FBXL8 knocked down BRCA cell lines was significantly reduced as shown by: (**C**) MTT and (**D**) CTB assays. For each time point, cell counts were normalized to the corresponding PBS-treated controls. (**E**) Alamar Blue or (**F**) Trypan Blue exclusion tests showed significant reduction in cell proliferation. (**G**) Annexin V and 7-aminoactinomycin (7-AAD) double staining was conducted to examine early apoptosis (AnnexinV^+^/7AAD^−^, red box). (**H**) FBXL8-siRNA knockdown of MCF7 and MDA-MB231 cells underwent apoptosis via activation of Caspases -9 and -3, suggesting that presence of FBXL8 is pivotal to anti-apoptosis in BRCA. Representative histograms of apoptosis assay are shown in Appendix A. Data are means ± SD (*n* = 3). ***, *p* < 0.001.

**Figure 4 cancers-12-02210-f004:**
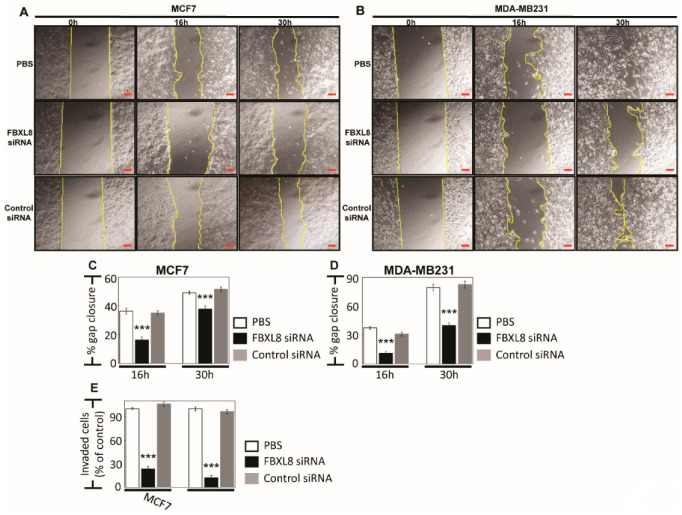
Knockdown of FBXL8 effectively suppressed BRCA migration and invasion. Cell migration assay was performed for: (**A**) MCF7 and (**B**) MDA-MB231 cells. Corresponding quantitative results are shown in (**C**) and (**D**). Compared with control siRNA, FBXL8 siRNA-treated cells showed significant decrease in the rate of migration. For each treatment, the migration rate (% gap closure) was normalized to the corresponding 0 h time point controls. Cell invasion assay and corresponding quantitative results are shown in Appendix A and (**E**), respectively. We showed that FBXL8-knockdown effectively reduced cell invasion. Data are representative of means ± SD (*n* = 3). Scale bar is 100 μm, shown as the red color line (—). ***, *p* < 0.001.

**Figure 5 cancers-12-02210-f005:**
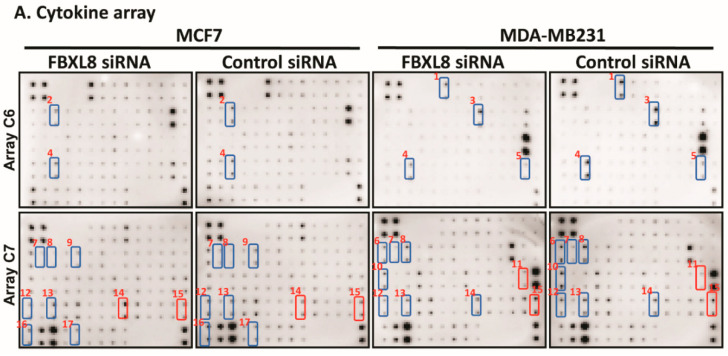
Knockdown of FBXL8 suppressed the production of inflammatory cytokines, chemokines and growth factors in breast carcinoma cells. Culture supernatants collected from cells treated with FBXL8-specific RNAi for 48 h, were analyzed for secreted tumorigenic factors using human cytokine array and ELISA. (**A**) Compared to control siRNA, treatment of FBXL8-specific RNAi significantly reduced the production of cancer-promoting factors, including MCP-1, I-TAC, TECK, CTACK, MIF, GM-CSF, NT-3, FGF-6, angiogenin, EGFR, ICAM-1, Fas, DtK and TRALIR3. In contrast, FBXL8 knockdown elevated the secretion of pro-tumorigenic IL-6R and TNFR1 in BRCA. Osteoprotegerin (OPG) shows opposite trend in MCF7 and MDA-MB231 cells. Blue boxes: down-regulation and red boxes: up-regulation of factors secreted by FBXL8-knockdowned cells. The numbers next to the blue/red boxes indicate: (1) Angiogenin, (2) FGF-6, (3) GM-CSF, (4) MCP-1, (5) NT-3, (6) CTACK, (7) DtK, (8) EGFR, (9) Fas, (10) ICAM-1, (11) IL-6R, (12) I-TAC, (13) MIF, (14) OPG, (15) TNFR1, (16) TECK and (17) TRALIR3. The array membranes are spotted in duplicate. (**B**) ELISA confirmed differential quantities of GM-CSF, MCP-1, CTACK, EGFR, ICAM-1, I-TAC, MIF and TECK secreted from MCF7 and MDA-MB231 cells when FBXL8 was knocked down (blue bars). Data are representative of means ± SD (*n* = 3). **, *p* < 0.01, ***, *p* < 0.001.

**Figure 6 cancers-12-02210-f006:**
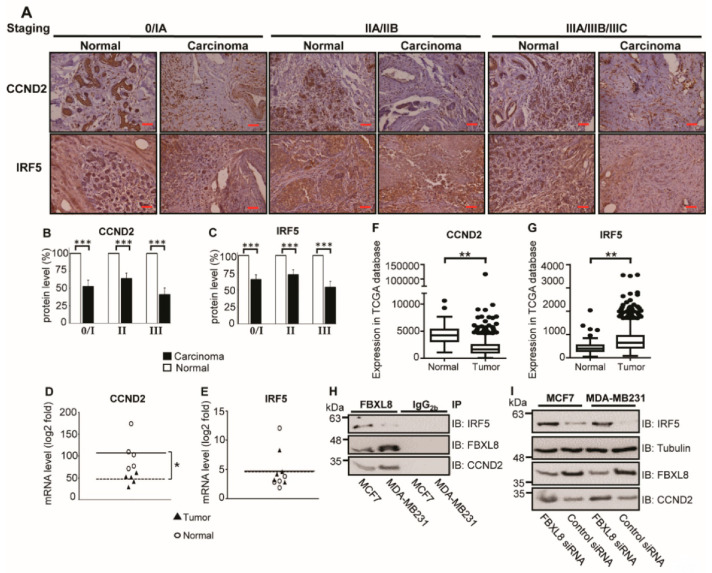
FBXL8 pulled down CCND2 and IRF5, and knockdown of FBXL8 accumulated CCND2 and IRF5 protein levels in BRCA cells. To clarify the potential of CCND2 and IRF5 in primary BRCA tissues, retrospective analyses were performed by IHC (A–C) and NGS-based RNA-Seq (D,E), and from TCGA database (F,G). (**A**) immunofluorescence staining of CCND2 and IRF5. Protein expression levels of CCND2 and IRF5 are examined in both normal (*n* = 30) and carcinoma (*n* = 30) breast tissues. The corresponding quantitative results of CCND2 and IRF5 in IHC are shown in panels (**B**,**C**), respectively. (**D**,**E**) Quantitative results from each patient. Each data point represents a log2 fold change value for CCND2 and IRF5 (*, FDR < 0.05). (**F**,**G**) Box-whisker plot showing the transcript levels of CCND2 and IRF5 in TCGA database, respectively. Vertical lines, box and horizontal white line correspond to min-max range, 25–75th percentile range and median, respectively. Data are reported as normalized count as provided in the TCGA level 3 data (**, *p* < 0.01). The corresponding clinicopathological information for IHC are shown in Appendix A. Brown color indicates DAB dye-stained protein of interest. Scale bar is 100 μm, shown as the red color line (—). (**H**) Co-IP of of FBXL8 and CCND2 (or IRF5) in BRCA cells; total cell lysates were immunoprecipitated with anti-FBXL8 (or control anti-IgG_2b_) antibody, separated by 10% SDS-PAGE (sodium dodecyl sulfate polyacrylamide gel electrophoresis), immunoblotted and probed with FBXL8, CCND2 or IRF5 antibodies. (**I**) To confirm whether CCND2 and IRF5 are regulated by FBXL8-dependent protein degradation, FBXL8 RNAi was performed in MCF7 and MDA-MB231 cells, followed by immunoblotting analysis. Tubulin was used as a loading control for immunoblotting.

**Figure 7 cancers-12-02210-f007:**
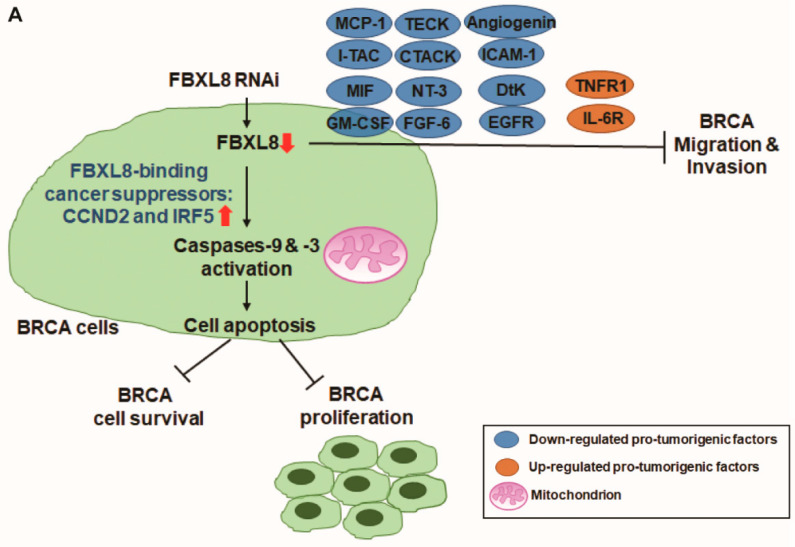
A hypothetical model illustrating how FBXL8 attenuates anti-cancer CCND2 and IRF5 factors, upregulates pro-tumorigenic cytokines, suppresses apoptosis and promotes metastasis potential in BRCA. FBXL8, a novel SCF E3 ubiquitin ligase, plays a key role in anti-apoptosis in BRCA. (**A**) Knockdown of FBXL8 increased expression of cancer suppressors CCND2 and IRF5, thus inhibiting cell growth and proliferation in BRCA, driven by: (i) increase in early apoptosis, (ii) activation of Caspase-9 and -3, and (iii) inhibition of the production of cancer-promoting cytokines, including MCP-1, I-TAC, TECK, CTACK, MIF, GM-CSF, NT-3, FGF-6, Angiogenin, ICAM-1, DtK and EGFR. Therefore, FBXL8 promotes pro-tumorigenic microenvironment, contributing to BRCA metastasis and progression. (**B**) FBXL8 interacts and downregulates cancer suppressors CCND2 and IRF5 via protein degradation system. Hence, upregulated FBXL8 in BRCA causes the reduction of CCND2 and IRF5 proteins, and leads to dysregulated cell proliferation, immune response and metastatic potential of BRCA cells.

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
