# Peer review of "Human FBXL8 Is a Novel E3 Ligase Which Promotes BRCA Metastasis by Stimulating Pro-Tumorigenic Cytokines and Inhibiting Tumor Suppressors"

_cancers, 2020, doi:10.3390/cancers12082210_

Round 1
Reviewer 1 Report
In this study, the authors investigate the role of FBXL8 in breast cancer. The study is interesting. Some sentences read as overstatements. A couple of additional experiments will also likely allow the authors to reach their current conclusions as detailed below.
Abstract:
Line 28 « which is subtly balanced by the E3 ligase » should be rephrased, as this reads unspecific.
« the pivotal E3-ligase in BRCA »: this is an overstatement
Line 40: this is an overstatement, this could only be shown if in vivo experiments were performed, which is not the case in this study. Same comment for BRCA progression line 42, this needs to be toned down.
Line 55: Is this currently tested for breast cancer? The corresponding clinical trial number(s) should be added.
Line 66: there’s a typo for Bortezomib
Part 2.2: the authors should quickly explain why they discarded exploring FBXW7 and FBXO4 (and/or why they decided to focus on FBXL8).
Line 166 « The rise in the ubiquitin protein in BRCA tissues (Figure 2E) corroborates the significance of the UPS in BRCA development… » This needs to be slightly rephrased, ubiquitination is not necessarily associated with involvement of the proteasome and proteasomal-dependent degradation.
Figure 3A: Is FBXL8 also different at the protein level in those cell lines?
Figure 3B: Can the authors provide the raw data for both graphs? These seems surprisingly identical.
Figure 5: Some of the factors regulated are transmembrane proteins. How do the author explain their presence in culture supernatant? Was it reported previously? Are these factors also regulated intracellularly, at the mRNA level?
Line 313: The authors need to clearly explain how they picked CCND2 and IRF5 for further investigation.
Figure 6H: corresponding lysates need to be shown. Conclusion is incorrect: the fact that FBXL8 and CCND2 (/IRF5) are part of a protein complex does not mean that CCNDS and IRF5 are substrates of FBXL8. This needs to be corrected.
Figure 6I: Does knock-down of FBXL8 impact on mRNA levels of CCND2 and/or IRF5? Or on its translation? In other words: how can the authors suggest that FBXL8 impacts on proteasomal degradation of these proteins?
In the abstract, the sentence « FBXL8 specifically interacts with and apparently degrades two tumor-suppressors, CCND2 and IRF5, to promote BRCA progression. » is not supported by any data, please rephrase.
Reviewer 2 Report
I suggest to avoid the list of methods used in abstract.
The number of samples used in figure 1A is very low and authors compared different type of cancers (different gene espression profile, grade and etc.).
Lines 122-125 : sentence is elusive and it is not corroborated by evidence on “its fundamental biological Roles”.
What type of tumors samples in TGCA authors analyzed?
A western blot analysis revealing the FBXL8 silencing is lacking.
In figure 2 authors assessed that the expression level of FBXL8 increases with the stage of BRCA. However, it seems that the result doesn’t agree with those showed in figure 1.
Figure 4: it is very strange that cells can survive and migrate in PBS until 30h.
EGFR is not a cyto/chemokines.
As elsewhere reported, MDA-MB231 is often used as a triple negative breast cancer cell line.
10.3389/fendo.2018.00492
In Figure 5 authors show that in basal condition MDA expresses EGFR. Please explain the concern.
How authors explain the same behaviour of MCF7 and MDA-MB231 cell Lines?
Why authors do not show results obtained in MCF10 cell line?
A preprint of the manuscript is available at
please give an explanation.
Round 2
Reviewer 1 Report
In the revised version of the manuscript, the authors have satisfactorily addressed my previous comments. I do not have any further questions.
Reviewer 2 Report
Authors addressed my suggestions. I accept the revised manuscript.